# Fine-Tuning of mTORC1-ULK1-PP2A Regulatory Triangle Is Crucial for Robust Autophagic Response upon Cellular Stress

**DOI:** 10.3390/biom12111587

**Published:** 2022-10-28

**Authors:** Bence Hajdú, Marianna Holczer, Gergely Horváth, Gábor Szederkényi, Orsolya Kapuy

**Affiliations:** 1Department of Molecular Biology, Institute of Biochemistry and Molecular Biology, Semmelweis University, 1083 Budapest, Hungary; 2Faculty of Information Technology and Bionics, Pázmány Péter Catholic University, 1083 Budapest, Hungary

**Keywords:** mTORC1, PP2A, ULK1, autophagy, systems biology, feedback loop

## Abstract

Autophagy-dependent cellular survival is tightly regulated by both kinases and phosphatases. While mTORC1 inhibits autophagy by phosphorylating ULK1, PP2A is able to remove this phosphate group from ULK1 and promotes the key inducer of autophagosome formation. However, ULK1 inhibits mTORC1, mTORC1 is able to down-regulate PP2A. In addition, the active ULK1 promotes PP2A via phosphorylation. We claim that these double-negative (mTORC1 –| PP2A –| mTORC1, mTORC1 –| ULK1 –| mTORC1) and positive (ULK1 -> PP2A -> ULK1) feedback loops are all necessary for the robust, irreversible decision making process between the autophagy and non-autophagy states. We approach our scientific analysis from a systems biological perspective by applying both theoretical and molecular biological techniques. For molecular biological experiments, HEK293T cell line is used, meanwhile the dynamical features of the regulatory network are described by mathematical modelling. In our study, we explore the dynamical characteristic of mTORC1-ULK1-PP2A regulatory triangle in detail supposing that the positive feedback loops are essential to manage a robust cellular answer upon various cellular stress events (such as mTORC1 inhibition, starvation, PP2A inhibition or ULK1 silencing). We confirm that active ULK1 can up-regulate PP2A when mTORC1 is inactivated. By using theoretical analysis, we explain the importance of cellular PP2A level in stress response mechanism. We proved both experimentally and theoretically that PP2A down-regulation (via addition of okadaic acid) might generate a periodic repeat of autophagy induction. Understanding how the regulation of the cell survival occurs with the precise molecular balance of ULK1-mTORC1-PP2A in autophagy, is highly relevant in several cellular stress-related diseases (such as neurodegenerative diseases or diabetes) and might help to promote advanced therapies in the near future, too.

## 1. Introduction

Macroautophagy (autophagy) is an evolutionarily conserved self-eating mechanism when the cell forms autophagosomes (double-membrane vesicles) around the components that must be digested [1,2,3]. Autophagy is active under physiological conditions at low levels and it is more intense during various stress conditions (i.e. starvation or endoplasmic reticulum stress) [4]. All previous scientific results suggest that autophagy plays a key role in cell survival by regulating and maintaining cellular homeostasis [3]. One of the most important elements of the autophagy machinery is unc51-like autophagy activating kinase 1/2 (ULK1/2) [2,5]. ULK1/2 forms a complex with ATG13, ATG101 and FIP200 to control the autophagy induction [5,6]. Mammalian target of rapamycin (mTOR) is a serine/threonine protein kinase and the master regulator of the cellular metabolism. mTOR is the catalytic subunit of mTORC1, which complex comprises still mammalian lethal with SEC13 protein 8 (mLST8), PRAS40 and regulatory-associated protein of mTOR (Raptor). This complex controls among others the cell growth and division, proliferation, and autophagy, too [7,8]. Under physiological condition mTORC1 gets activated by a phosphatidylinositol 3 kinase (PI3K)-Akt kinase-dependent manner, however it is inhibited by hamartin-tuberin (TSC1/2) complex [9]. mTORC1 keeps autophagy in its inactive state through direct phosphorylation of AMP-activated protein kinase (AMPK) [10,11] and ULK1/2 under physiological conditions [5,12]. If mTORC1 gets inhibited, than the protein synthesis becomes blocked by the de-phosphorylation of ribosomal protein S6 kinase (p70S6K1) and translation initiation factor 4E binding protein-1 (4E-BP1), respectively [9].

One of the members of serine/threonine phosphatases, called protein phosphatase 2A (PP2A), plays an important role in maintaining cellular homeostasis by participating in the regulation of cell cycle, proliferation, cell death, and several signalling pathways [13,14]. PP2A is composed of a catalytic subunit (C subunit), a scaffold subunit (A subunit) and a highly variable regulatory subunit (B subunit) [13]. It has already proved that PP2A holoenzyme containing a B55-family regulatory subunit (PP2A-B55) regulates the autophagy through dephosphorylating of PHD2, ULK1 and Beclin1, the key inducers of self-cannibalism [15]. PP2A dephosphorylates PHD2 under hypoxia, reducing its activity and thus causing activation of autophagy via HIF1/2-Redd1-TSC1/2 pathway [15,16].

The proper balance of reversible phosphorylation/dephosphorylation of proteins is an essential regulatory mechanism in many cellular processes [17]. Namely, mTORC1 is able to phosphorylate ULK1 at both Ser638 and Ser757 residues under nutrient rich condition [5,12,18]. However, during starvation, PP2A can dephosphorylate ULK1 at Ser637, thereby promoting autophagy activation [15,19]. The active ULK1 kinase inhibits mTORC1 by phosphorylating its Raptor subunit [5,20,21]. This ULK1-dependent negative effect on mTORC1 creates a double negative feedback loop between mTORC1 and ULK1 [5]. PP2A inhibits the mTORC1 activation by regulating Akt with dephosphorylating at the Thr308 residues [22,23]. mTOR kinase phosphorylates and thus inhibits PP2A [24,25,26]. So, there is a double negative feedback loop between mTORC1 and PP2A [23,27], too. Furthermore, ULK1 may positively affect PP2A by phosphorylating striatin among its regulatory subunits, thus creating a positive feedback loop between ULK1 and PP2A [28].

In addition to various natural environmental influences (such as starvation), the balance of phosphorylation/dephosphorylation of autophagy-regulating proteins can be disturbed by the addition of various artificial substances. For example, rapamycin inhibits mTORC1 via forming a functional complex with 12-kDa FK506-binding protein (FKBP12) and binding to the mTOR subunit of mTORC1. In this case rapamycin acts as an allosteric inhibitor, thereby inhibiting the activation of p70S6K1 [29] and promoting the activation of ULK1 and autophagy can be initiated in the cell [30]. The well-known drug to inhibit the cellular dephosphorylation, called okadaic acid (OA). OA is a long-standing inhibitor of PP1 and PP2A phosphatases, but it has a significantly higher affinity for PP2A [31].

In this study we reveal an interesting dynamical characteristic of either mTORC1 inhibition or PP2A down-regulation treatment in human cell line. Namely, we suggest that the double negative and positive feedback loops of mTORC1-ULK1-PP2A regulatory triangle guarantees a robust autophagy induction upon cellular stress. Besides, mTORC1 down-regulation or PP2A inhibition might be able to generate an autophagy induction repeated periodically in time. By using both theoretical and molecular biological techniques we test the importance of each element of this above mentioned regulatory triangle and we also analyse the intensity of autophagy oscillation upon rapamycin treatment or addition of OA.

## 2. Materials and Methods

### 2.1. Materials

Rapamycin (Sigma-Aldrich, St. Louis, MO, USA; R0395), okadaic acid (Sigma-Aldrich, 495604), Bafilomycin A1 (Sigma-Aldrich, M17931), DMEM—no glucose, no glutamine (Life Technologies, Carlsbad, CA, USA; A14430-01) were purchased. All other chemicals were of reagent grade.

### 2.2. Cell Culture and Maintenance

As model system, human embryonic kidney (HEK293T, ATCC, Manassas, VA, USA; CRL-3216) cell line was used. It was maintained in DMEM (Life Technologies, 41965039) medium supplemented with 10% fetal bovine serum (Life Technologies, 10500064) and 1% antibiotics/antimycotics (Life Technologies, 15240062). Culture dishes and cell treatment plates were kept in a humidified incubator at 37 °C in 95% air and 5% CO${}_{2}$.

### 2.3. SDS-PAGE and Western Blot Analysis

Cells were harvested and lysed with 20 mM Tris, 135 mM NaCl, 10% glycerol, 1% NP40, pH 6.8. Protein content of cell lysates was measured using Pierce BCA Protein Assay (Thermo Scientific, Waltham, MA, USA; 23225). During each procedure equal amounts of protein were used. SDS-PAGE was done by using Hoefer miniVE (Amersham, UK). Proteins were transferred onto Millipore 0.45 µM PVDF membrane. Immunoblotting was performed using TBS Tween (0.1%), containing 5% non-fat dry milk (Sigma-Aldrich, 70166) or 1% bovine serum albumin (Sigma-Aldrich, A9647) for blocking membrane and for antibody solutions. Loading was controlled by developing membranes for GAPDH in each experiment. For each experiment at least three independent measurements were carried out. The following antibodies were applied: antiLC3B (Santa Cruz Biotechnology, Dallas, TX, USA; sc-271625), antip62 (Cell Signaling Technology, Danvers, MA, USA; 5114S), antiULK1-Ser757-P (Cell Signaling Technology, 6888S), antiULK1 (Cell Signaling Technology, 8054S), antip70S6K-P (Cell Signaling Technology, 9234S), antip70S6K (Santa Cruz, sc-9202), antiAMPK-P (Cell Signaling Technology, 2531S), antiAMPK (Cell Signaling Technology, 2603S), antiPP2A-Tyr307-P (Sigma-Aldrich, SAB4503975), antiPP2A C Subunit (Cell Signaling Technology, 2259S) and antiGAPDH (Santa Cruz, 6C5), HRP conjugated secondary antibodies (Cell Signaling Technology, 7074S, 7076S). The bands were visualised using chemiluminescence detection kit (Thermo Scientific, 32106).

### 2.4. Silencing with siRNA

Cells were harvested and then seeded in six-well plates (200,000 cells/well) in antibiotic-free medium. Cells were allowed to settle overnight and transfected the next day. Lipofectamine RNAi Max (Invitrogen, Waltham, MA, USA; 13778075) reagent, GIBCO™Opti-MEM I (GlutaMAX™-I) reduced serum medium (Invitrogen, 31985070) and siRNA at a concentration of 20 pmol/ml were used for transfection. The ULK1 and PP2ACα siRNAs were purchased from Ambion (118259, 104510, s10957, s10958). The reagent was added to the cells and incubated for 24 h, followed by the treatments. Silencing efficiency was checked at protein levels.

### 2.5. Mathematical Modelling

The regulatory network was translated into a set of nonlinear ordinary differential equations (ODEs) and analysed using the techniques of dynamical system theory [32,33,34]. For details see Appendix A. All the used codes are available on a code hosting platform for future collaborations, called GitHub: https://github.com/eraut/pp2aMtorUlk, accessed on 28 August 2022.

### 2.6. Statistics

For densitometry analysis Western blot data were acquired using ImageJ software. For the phosphorylated forms of p70S6K, ULK1, PP2A, and AMPK, relative band densities were normalized to the corresponding total protein, while LC3 II and p62 proteins’ relative band densities were normalized to GAPDH. Then the treated data series were normalized for each protein with its own control. For each of the experiments three independent measurements were carried out. Results are presented as mean values ± S.D. and were compared using *t*-Test (two sample assuming unequal variances) with Bonferroni correction (*p*-value correction). Asterisks indicate statistically significant difference from the appropriate control: ns—nonsignificant; *—*p* < 0.05; **—*p* < 0.01.

The quantitative analysis of the Western blot measurements was repeated with Azure-Spot Pro software to check the results acquired with ImageJ. The program contains 6 different background deduction method, we used the rolling ball method with a constant radius of 2 units. The different bands then were detected with the software’s default setting, false detections were corrected by hand.

## 3. Results

### 3.1. Building Up the Simple Mathematical Model of ULK1/2 Phosphorylation/Dephosphorylation

To investigate the system-level importance of the feedback loops of the regulatory network of mTORC1-ULK1-PP2A (referring to mTORC1, ULK1/2 and PP2A/B55, respectively) upon cellular stress both molecular and theoretical biological techniques were used. First a wiring diagram of the simple model was generated (Figure 1).

It is well-known that there is a double negative feedback loop between mTORC1 and ULK1 kinases (mTORC1 -| ULK1 -| mTORC1). Namely not only the key integrator of intracellular sensing is able to down-regulate the main subunit of autophagy activator complex, but the inducer of the self-digesting process also has a negative effect on mTORC1 via inhibitory phosphorylation. (see regulatory connections ‘a’ and ‘b’ on Figure 1) [5,12,18,20,21]. It is also proved that PP2A phosphatase is able to remove an inhibitory phosphate group from ULK1 promoting its activation [15]. Recently published data are confirmed that ULK1 is also able to induce PP2A via phosphorylation suggesting a positive feedback loop (i.e., ULK1 -> PP2A -> ULK1) in the control network (see regulatory connections ‘c’ and ‘d’ on Figure 1) [28]. Interestingly, a PP2A-dependent inhibition of mTORC1 has been also observed, which has a negative effect on mTORC1 activity [22,23,35,36,37]. Besides, mTORC1 can inhibit PP2A via phosphorylation [24,25,26] generating a mutual antagonism between the two proteins (mTORC1 -| PP2A -| mTORC1) (see regulatory connections ‘e’ and ‘f’ on Figure 1).

To explore the dynamical characteristic of the control network ordinary differential equations were written for the activity change of each protein (i.e., ULK1, mTORC1 and PP2A) and the precise parameters of the equations were estimated directly based on our experimental data.

To find the proper parameters for computer simulations HEK293T cells were treated with mTORC1 inhibitor (100 nM rapamycin for 60 min, see Figure 2) or with PP2A inhibitor (100 nM OA for 60 min, see Figure 3). Samples were taken at specific time intervals (at every 10 min). The time-dependency of the key indicators of mTORC1 (phosphorylation of its target, p70S6K), ULK1 (phosphorylation status of Ser-757 residue of ULK1) and PP2A (phosphorylation status of PP2A) were detected by immunoblotting during treatments (Figure 2A and Figure 3A). mTORC1 quickly got down-regulated 10 min after rapamycin addition (see the dephosphorylation of p70S6K), while both ULK1 and PP2A became active after 30 min (see the dephosphorylation of both ULK1-757-P and PP2A-P) (Figure 2A,B).

In contrast, OA treatment resulted in a significant inhibition of PP2A (PP2A remained in its phosphorylated state on Figure 3A,B), meanwhile mTORC1 got hyper-activated. High level of mTORC1 was able to keep ULK1 in its inactive state, i.e., decrease in ULK1-757-P level was not observed.

Then a simple mechanistic model of the mTORC1-ULK1-PP2A regulatory triangle was built up, where mass action kinetics were used for sake of simplicity. The resulting model contained three ordinary differential equations and 12 unknown parameters (reaction rate constants). Since values of the model parameters are not possible to determine via experimental methods, we had to determine them from time series data with the use of an optimization algorithm. The optimization algorithm tries to minimize the value of a so-called cost function. The cost function quantifies the difference between our measurement data and the simulation results. Detailed model equations and the optimization algorithms used can be found in the Appendix A.

With our molecular biological analysis, we first built up a simple theoretical model of mTORC1-ULK1-PP2A regulatory triangle whose parameters were taken directly from our experimental results.

### 3.2. The Double Negative and Positive Feedback Loops of the Control Network Can Generate a Bistable Characteristic of Autophagy Induction

To explore the importance of positive and double negative feedback loops in the dynamical characteristic of mTORC1-ULK1-PP2A regulatory triangle, we plotted phase plane diagrams, where ordinary differential equations were written for the two “main” enemies, namely ULK1 and mTORC1, respectively (Figure 4).

In a coordinate system the so called balance curve of ULK1 (see green line on Figure 4) and mTORC1 (see orange line on Figure 4) were depicted. Where the balance curves intersect each other, the system might be in stable steady state. Due to the positive (between ULK1 and PP2A) and double negative (between ULK1 and mTORC1 and mTORC1 and PP2A) feedback loops in the control network the balance curved were waved and had three intersections, namely two stable states were separated with an unstable one under physiological conditions (see the black and white dots on Figure 4A). The two stable states referred to “non-autophagy state” (with active mTORC1 and inactive ULK1) and “autophagy state” (with active ULK1 and inactive mTORC1), respectively. Since originally the level of active mTORC1 was high, the system chose the physiological state with inactive autophagy (see black dot with “Non-aut.st.” on Figure 4A).

However, addition of rapamycin caused in a quick inactivation of mTORC1, which resulted in the mTORC1 balance curve shifting to the left (Figure 4B). In this case, the stable “Non-aut.st.” steady state disappeared and the control system was forced to “jump” to its only one equilibrium point, which corresponded to autophagy state (see black dot with “Aut.st.” on with active ULK1 and inactive mTORC1 Figure 4B).

Although the balance curve of ULK1 shifted to the left in case of OA treatment, both steady states were still present (i.e., “Non-aut.st.” and “Aut.st.” on Figure 4C). Since mTORC1 was high under physiological conditions, the system remained in this state (see black dot with “Non-aut.st.” on Figure 4C). Due to the hyper-active mTORC1 and the absence of PP2A, ULK1 did not get active either.

Phase plane analysis of the mTORC1-ULK1-PP2A regulatory triangle has confirmed the presence of bistability in the control network and whether the non-autophagy or autophagy states appeared depended largely on the treatment.

### 3.3. The Outcome of mTORC1 Inhibition Highly Depends on the Cellular Level of Either ULK1 or PP2A

To further investigate the feedback loops in the control network ULK1 or PP2A down-regulation was combined with mTORC1 inhibition and the dynamical characteristic of the regulatory network was followed by both experimentally and theoretically (Figure 5 and Figure 6).

To investigate the role of ULK1 in this mTORC1-ULK1-PP2A regulatory triangle ULK1 was diminished with ULK1 siRNA transfection, meanwhile mTORC1 activity was inhibited by rapamycin treatment (100 nM for 2 h) in HEK293T cells. The characteristic of the key indicators of mTORC1 (phosphorylation of its target, p70S6K), ULK1 (phosphorylation status of Ser-757 residue of ULK1) and PP2A (phosphorylation status of PP2A) were detected by immunoblotting during end of treatments (see Figure 5). Although mTORC1 activity was completely decreased (see the dephosphorylation of p70S6K), in the absence of ULK1 most of the cellular PP2A remained inactive, i.e., a high level of PP2A-P was observed which referred to its inactive form. With computer simulations besides mTORC1 inhibition, different levels of reduced ULK1 levels were tested (Figure 5C,D), but no significant activation of PP2A was observed in any of the cases suggesting the importance of ULK1 in activation of PP2A upon rapamycin treatment.

PP2A was diminished with either addition of OA (100 nM for 3 h, see Figure 6A) or transfection with PP2A siRNA (see Figure 6B), meanwhile mTORC1 activity was inhibited by rapamycin treatment (100 nM for 2 h) in HEK293T cells. The characteristic of the key indicators of mTORC1 (phosphorylation of its target, p70S6K), ULK1 (phosphorylation status of Ser-757 residue of ULK1) and PP2A (phosphorylation status of PP2A) were detected by immunoblotting during end of treatments (Figure 6).

Since the ratio of PP2A-P/PP2A increased significantly during OA treatment, a total inhibition of the phosphatase was assumed (Figure 6A). Densitometry data confirmed a significant decrease of the level of PP2A in case of transfection with PP2A siRNA, although 25% of the original PP2A level was still present in the cells (Figure 6B).

OA treatment combined with rapamycin treatment resulted in ULK1 activation (see the remove of inhibitory phosphate residue at Ser-757 site from ULK1 on Figure 6A). In contrast, PP2A siRNA transfection combined with mTORC1 down-regulation was able to keep ULK1 in its inactive state by ULK1 remaining phosphorylated on its Ser-757 (Figure 6B). Our computational simulation results were fully consistent with the experimental data when mTORC1 inhibition was combined with full (equivalent to silencing with PP2A siRNA) or partial (equivalent to OA treatment) PP2A inhibition (Figure 6C,D). Similar results were observed when mTORC1 was down-regulated by glucose starvation in HEK293T cells (see Appendix A).

Taken together these results, we have demonstrated for the first time the role of both PP2A and ULK1 upon mTORC1 inhibition.

### 3.4. PP2A Inhibition with OA Might Generate a Periodic Repeat of Autophagy

Recently, we have published a systems biological analysis, where we could verify both experimentally and theoretically the oscillatory characteristics of the mTORC1-AMPK-ULK1 regulatory triangle under mTORC1 inhibition. Namely, the ULK1-AMPK-P negative feedback exhibited limit cycle oscillation upon prolonged rapamycin treatment in HEK293T cells [38]. Here we confirmed that PP2A-P also depicted periodic phosphorylation, when HEK293T cells were treated with 100 nM rapamycin for 120 min and samples were taken in every 15 min (see Appendix A). PP2A was always activated together with ULK1 (and the autophagy markers), while mTORC1 became active in the next wave when PP2A and ULK1 were inactivated, while the whole process was repeated with a one-hour period.

Question immediately arose, whether prolonged PP2A inhibition could result in periodic repeat of mTORC1 and ULK1. Therefore, HEK293T cells were treated with 100 nM or 175 nM OA for 180 min, and samples were taken in every 30 min (Figure 7). The characteristic of the key indicators of mTORC1 (phosphorylation of its target, p70S6K), ULK1 (phosphorylation status of Ser-757 residue of ULK1) and PP2A (phosphorylation status of PP2A) were detected and the key markers of autophagy (i.e., p62 level and the ratio of LC3II/LC3I) were depicted by immunoblotting during end of treatments (Figure 7). Interestingly, the activation of the proteins has shown a rhythmic pattern having a period of approximately 1.5 h upon 100 nM OA treatment (Figure 7A). The activation profiles of p70S6K-P, PP2A-P and ULK1-P changed periodically. When PP2A and ULK1 seemed to be active (see their dephosphorylation on Figure 7A), p62 level got decreased and LC3II/LC3I level got increased assuming a periodic activation of autophagy. mTORC1 always got activated when the autophagy supposed to be inactive. To rule out that the periodic change of autophagy could be due to cell cycle or circadian rhythm, therefore OA treatment was carried out in non-synchronized cell population.

Interestingly, in case of prolonged treatment with even higher level of OA (175 nM for 180 min) no periodic repeat of either mTORC1 or ULK1 was observed (Figure 7B), suggesting that this above mentioned characteristic feature of the control network was highly dependent on the level of PP2A inhibition. To further confirm that autophagy induction has been properly achieved, the above mentioned OA treatments were combined with a well-known autophagy inhibitor, called Bafilomycin A1 (Appendix A). When autophagy flux was inhibited, higher levels of LC3II/GAPDH and p62/GAPDH were obtained, confirming that autophagy functioned properly during pure OA treatment (Appendix A).

With our data we first demonstrate that the periodic activation of autophagy was clearly observed in case of PP2A inhibition with OA.

## 4. Discussion

Maintaining cellular homeostasis against external and/or internal stimuli is one of the main goals of the living organism. The proper response of cells to both the anabolic metabolites around them and to adverse cellular stresses (such as oxidative stress or starvation) is essential for survival. By using systems biological techniques, recently we have shown the dynamical features of mTORC1-AMPK-ULK1 regulatory triangle induced autophagy-dependent survival upon various cellular stress events [10,38,39]. We claim that the feedback loops between these three kinases are essential for the proper induction of autophagy, where rapid phosphorylation steps are able to generate a rapid response. Besides, regulated kinase-dependent phosphorylation, addition of a regulated phosphatase can generate an even more robust response to the stimulus [17]. Therefore, for a proper balance of reversible phosphorylation/dephosphorylation in the signal transduction pathway, the role of phosphatases should not be forgotten.

Recently it has shown that serine/threonine phosphatase, called protein phosphatase 2A (PP2A) with its B55-family regulatory subunit is able to promote autophagy induction via dephosphorylation of ULK1, the key inducer of self-cannibalism [19]. This inhibitory phosphate group is placed on ULK1 by mTORC1 under physiological conditions, thus inhibiting autophagy induction [5,12]. It is well-known that ULK1 has a negative effect on mTORC1 generating a double negative feedback loop in the control network [5,20,21]. Besides, ULK1 enhances PP2A activity via phosphorylation, which creates a positive feedback loop (i.e., ULK1 -> PP2A -> ULK1) [28]. In addition, a mutual antagonism is observed between mTORC1 kinase and PP2A phosphatase, suggesting another double negative feedback loop in the control network [22,23,24,25,35,36,37] (Figure 1). Here we explored in the first time the dynamical characteristic of mTORC1-ULK1-PP2A controlled reversible phosphorylation/dephosphorylation upon cellular stress by approaching our analysis from a systems biological perspective.

To find valid parameters for our computer simulations both mTORC1 inhibition (via addition of rapamycin) and PP2A inhibition (via addition of OA) were performed in HEK293T cells, then the parameters of the model were fitted according to the densitometry data of active form of each molecules (i.e., mTORC1 marker p70S6K, PP2A and ULK1) (Figure 2 and Figure 3). This is the first time when a mathematical model was constructed and both regulated kinase (i.e., mTORC1) and regulated phosphatase (i.e., PP2A) were considered in the regulation of ULK1, and our own experimental results were used to determine the appropriate parameters of our simple model.

Here, we first tried to obtain the parameters for our model by using densitometric values of western blot results from experiments. It was particularly important that our densitometric results were the most accurate, so we used two programs for the analysis (i.e., imageJ and Azure). It can be said that the results were largely the same, but sometimes there were variations in standard deviations. This was particularly noticeable when trying to compare two “columns” with relatively similar characteristics. In this case, the amount of variance calculated by the program largely determined whether the difference became significant at all, and if so, to what extent. For example, this was also the case when mTOR was minimally activated by OA treatment (Figure 3B). During treatment, phosphorylation of ULK-757 increased minimally, which was not significant according to ImageJ (Figure 3B), whereas it became significant according to Azure (Appendix A). This discrepancy did not affect the fact, which is clear from the blots, that mTOR did indeed remain active. In all cases, however, we carefully checked that the results of the densitometry were correct.

To investigate the dynamical characteristic of mTORC1-ULK1-PP2A regulatory network upon cellular stress signal-response curves were generated (Figure 4). Although this model contains only three components but already three positive feedback loops exist, namely two double negative feedback loops (mTORC1 -| PP2A -| mTORC1; mTORC1 -| ULK1 -| mTORC1) and one positive feedback loop (ULK1 -> PP2A -> ULK1). Positive feedback loops make possible to generate a bistable system with two well-separated stable states. In this case, under physiological conditions mTORC1 promotes translation meanwhile ULK1 is inactive (see “Non-aut.st.” on Figure 4A), while the lower steady state quickly disappears if mTORC1 gets inhibited with rapamycin (see “Aut.st.” on Figure 4B). At the same time, PP2A inhibition via addition of OA does not affect the number of steady states and mTORC1 remains active (Figure 4C). It should be noted here that although both steady states (see “Non-aut.st.” and “Aut.st.” on Figure 4A) can “technically” be observed under physiological conditions, since in this case mTORC1 is active, the cell is in “Non-aut.st.” anyway.

We also suggest that these feedback loops have an essential role in guarantying robust autophagic response upon cellular stress. Our dynamical analysis shows that, due to the interacting feedback loops, an appropriate cellular stress response can be generated even if a failure causes one of the feedback loops to weaken or even drop out of the regulatory network completely, the bistable autophagy response might not be affected (data not shown).

To further investigate the importance of feedback loops “combined inhibitions” were performed, namely besides rapamycin-dependent mTORC1 inhibition either PP2A or ULK1 was down-regulated (Figure 5 and Figure 6). PP2A was diminished with either addition of OA or transfection with PP2A siRNA, while ULK1 inhibition was achieved with ULK1 siRNA, only. Interestingly, silencing with PP2A siRNA or addition of OA had different outcome in cases where mTORC1 was inhibited in the cells. Silencing with PP2A siRNA seems more drastic than OA treatment (Appendix A). Namely, ULK1 could be activated during OA treatment, while in the second case it remained completely inactive (Figure 6 and Appendix A). It is possible that some PP2A activity may have remained in the cell during our treatment with OA. However, silencing of PP2A with siRNA might achieve a much more precise PP2A inhibition in the cell. While in the first case, the small amount of PP2A can activate ULK1, during PP2A silencing, even if mTORC1 is inhibited by rapamycin, ULK1 remains inactive.

Similar to our recently published data [38] here we confirmed a sustained oscillatory characteristic of autophagy induction upon prolonged rapamycin treatment (Appendix A). In the absence of mTORC1 the negative feedback loop between AMPK and ULK1 was able to generate the periodic re-activation of self-cannibalism [38]. We assumed that with this periodic change of autophagy the cellular system might have an opportunity to utilize the building blocks produced during the digestion process. Question immediately arises whether the down-regulation of the phosphatase is able to generate similar characteristic of autophagy activity. An intermediate level of prolonged OA treatment was able to achieve a periodic activation of all the three molecules of the mTORC1-ULK1-PP2A regulatory triangle resulting in the periodic activation of autophagy induction, too (Figure 7A).

We observed that mTORC1-ULK1-PP2A regulatory triangle does not contain any negative feedback loops at all, which is essential for a sustained oscillatory mechanism. Therefore, we claim that in order for the model to describe the periodic induction of autophagy, it is necessary to incorporate extra links and/or extra elements in the control network. It is a possibility that AMPK has an important role in generating negative feedback loop in the control network. It is well-known that there is a negative feedback loop between AMPK and ULK1, AMPK–ULK1–mTORC1 triangle also generates a negative circuit in the regulatory network. In addition, the link between PP2A and AMPK may also contain negative and positive feedback loops [40,41]. Exactly what plays a role in the oscillation of autophagy during OA treatment remains to be investigated.

Interestingly, a more drastic concentration of OA totally blocked autophagy induction in the cell (Figure 7B). Since high level of OA may inhibit other phosphatases besides PP2A (such as PP1), we can assume that the absence of these phosphatases somehow inhibits the negative feedback loop that is essential for periodic induction of autophagy. Our results suggest that not only PP2A but also other phosphatases may play an important role in the stress response. However, this will need to be experimentally demonstrated in the future.

We must always bear in mind that autophagy dysfunction is associated with many serious diseases such as neurodegenerative diseases (e.g., Parkinson’s disease, Alzheimer’s disease), metabolic diseases, carcinogenesis and inflammatory bowel diseases (such as ulcerative colitis (UC) and Crohn’s disease (CD) [42]. It has shown that autophagy induction was beneficial in case of UC, although hyper-activation of autophagy could result in cell death [43,44]. Other findings also suggest that PP2A inhibition via mesalazine might be therapeutically important in colorectal cancer [45]. Combining these results with our observations, we conclude that a treatment with an appropriately adjusted OA concentration, in which autophagy is periodically repeated, might be a very promising treatment for UC.

Although our model is very simple, later we can build more sophisticated models by progressively extending this model, therefore this study definitively has medical importance in the future. Our systems biological approach improves the understanding of the molecular basis of these complex syndromes and might help to promote future therapeutic technics against the above mentioned diseases, too.

## Figures and Tables

**Figure 1 biomolecules-12-01587-f001:**
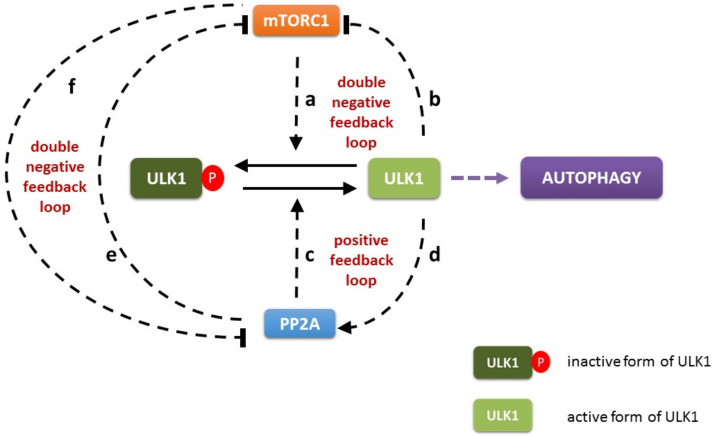
The wiring diagram of PP2A-mTORC1-ULK1 regulatory triangle. mTORC1 can inhibit the activation of both ULK1 (a) and PP2A (f) by phosphorylation. In contrast, active ULK1 inhibits mTORC1 (b) and stimulates PP2A (d) by phosphorylation. PP2A can abolish the inhibition of ULK1 by dephosphorylation (c) and prevent mTORC1 activation (e). The PP2A, mTORC1 and ULK1 are denoted by isolated blue, red and green boxes, respectively. Dashed line shows how the components can influence each other, while solid black lines denote biochemical reaction.

**Figure 2 biomolecules-12-01587-f002:**
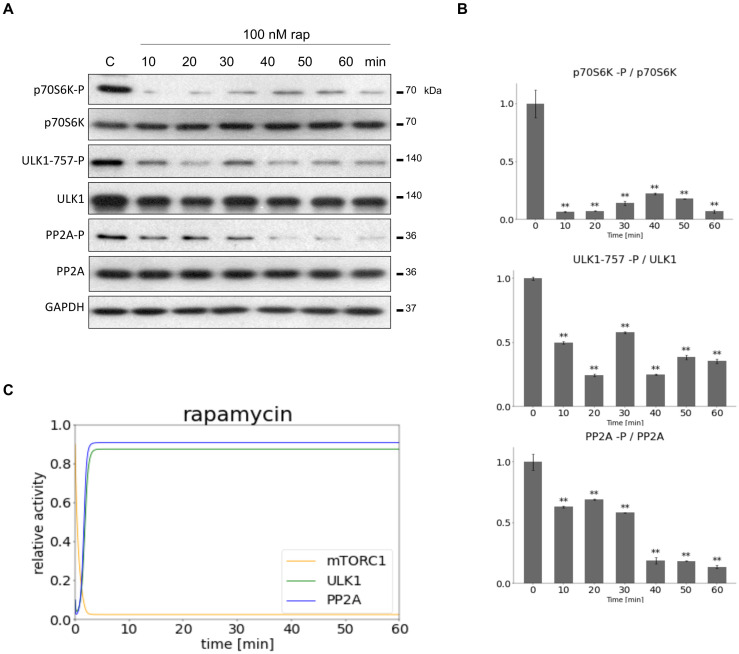
Inhibiting mTORC1 activity to set the parameters of our minimal model. HEK293T cells were denoted in time after 100 nM rapamycin treatment. (**A**) The markers of ULK1 (ULK1-757-P), PP2A (PP2A-P) and mTORC1 (p70S6K-P) were followed by immunoblotting. GAPDH was used as loading control. (**B**) Densitometry data represent the intensity of ULK1-757-P normalized for total level of ULK1, PP2A-P normalized for total level of PP2A and p70S6K-P normalized for total level of p70S6K. For each of the experiments, three independent measurements were carried out. Error bars represent standard deviation asterisks indicate statistically significant difference from the control: ns—nonsignificant; **—*p* < 0.01. (**C**) The computational simulation is determined upon rapamycin treatment (mTORT = 0.1). The relative activity of mTORC1, PP2A, ULK1 is shown.

**Figure 3 biomolecules-12-01587-f003:**
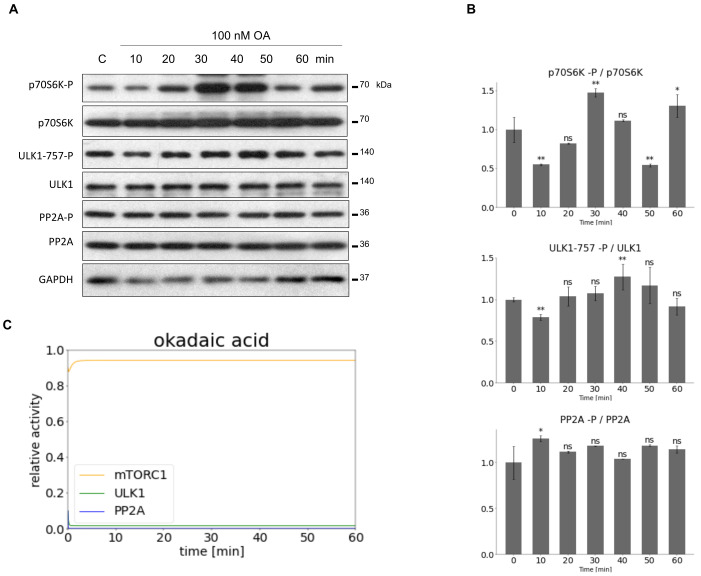
Inhibiting PP2A activity to set the parameters of our minimal model. HEK293T cells were denoted in time after 100 nM OA. (**A**) The markers of ULK1 (ULK1-757-P), PP2A (PP2A-P) and mTORC1 (p70S6K-P) were followed by immunoblotting. GAPDH was used as loading control. (**B**) Densitometry data represent the intensity of ULK1-757-P normalized for total level of ULK1, PP2A-P normalized for total level of PP2AC and p70S6K-P normalized for total level of p70S6K. For each of the experiments, three independent measurements were carried out. Error bars represent standard deviation asterisks indicate statistically significant difference from the control: ns—nonsignificant; *—*p* < 0.05; **—*p* < 0.01. (**C**) The computational simulation is determined upon OA treatment (PP2AT = 0.1). The relative activity of mTORC1, PP2A, ULK1 is shown.

**Figure 4 biomolecules-12-01587-f004:**
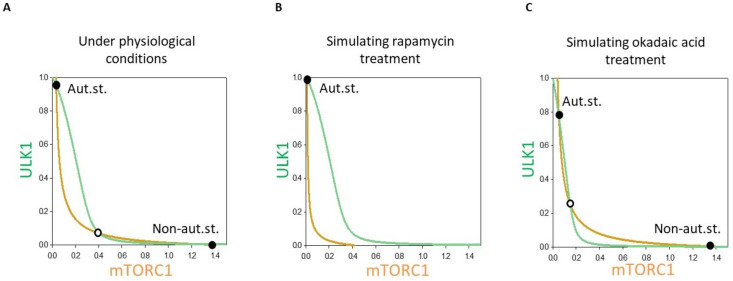
The dynamical feature of PP2A-mTORC1-ULK1 regulatory triangle controlled stress response mechanism. Phase plane diagram of PP2A-mTORC1-ULK1 regulatory triangle (**A**) under physiological conditions, upon (**B**) rapamycin (mTORT = 0.1) or (**C**) OA treatment (PP2AT = 0.3). The balance curves of ULK1 (green curve) and mTORC1 (orange curve) are plotted. Stable and unstable steady states are visualized with black and white dots, respectively. “Non-aut. state” refers to non-autophagy state (with active mTORC1 and inactive ULK1), while “Aut.st.” refers to “autophagy state” (with active ULK1 and inactive mTORC1).

**Figure 5 biomolecules-12-01587-f005:**
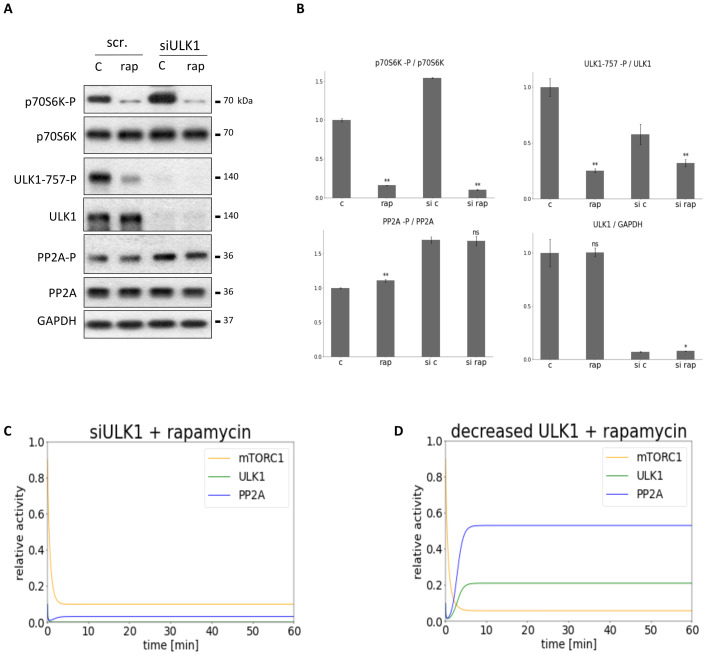
Combined down-regulation of mTORC1 and ULK1 can result in PP2A inhibition. ULK1 was silenced in HEK293T cells, then cells were treated with/without 100 nM rapamycin for 2 h. The silencing was checked by using a scramble siRNA. (**A**) The markers of ULK1 (ULK1-757-P), PP2A (PP2A-P) and mTORC1 (p70S6K-P) were followed by immunoblotting. GAPDH was used as loading control. (**B**) Densitometry data represent the intensity of ULK1-757-P normalized for total level of ULK1, PP2A-P normalized for total level of PP2A and p70S6K-P normalized for total level of p70S6K. For each of the experiments, three independent measurements were carried out. Error bars represent standard deviation asterisks indicate statistically significant difference from the control: ns—nonsignificant; *—*p* < 0.05; **—*p* < 0.01. The computational simulation is determined upon two different types of ULK1 silencing combined with rapamycin treatment: (**C**) (ULK1T = 0.01, mTORT = 0.1) and (**D**) (ULK1T = 0.3, mTORT = 0.1). The relative activity of mTORC1, PP2A, ULK1 is shown.

**Figure 6 biomolecules-12-01587-f006:**
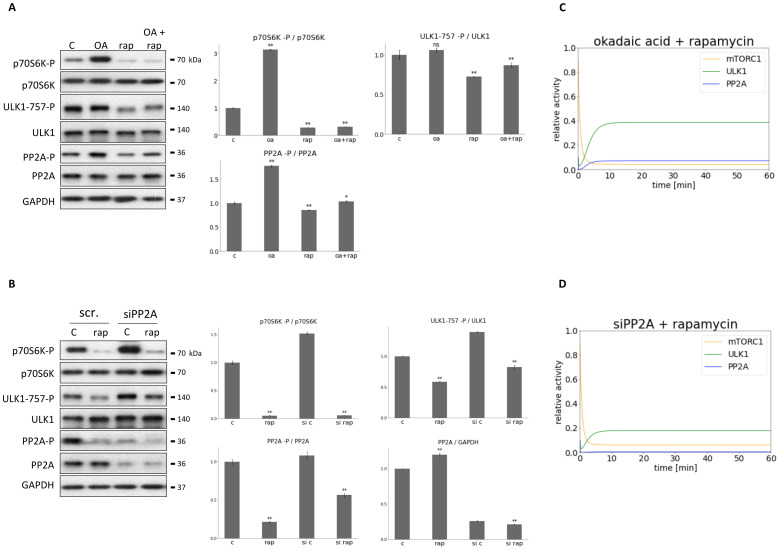
Combined down-regulation of both PP2A and mTORC1 can result in ULK1 activation. (**A**) HEK293T cells were treated with okadaic acid (OA—100 nM, 3 h), rapamycin (rap—100 nM, 2 h), or rap + OA. (**B**) PP2A was silenced in HEK293T cells, then cells were treated with/withouth 100 nM rap for 2 h. The silencing was checked by using a scramble siRNA. (**A**,**B**, **panel left**) The markers of ULK1 (ULK1-757-P), PP2A (PP2A-P) and mTORC1 (p70S6K-P) were followed by immunoblotting. GAPDH was used as loading control. (**A**,**B**, **panel right**) Densitometry data represent the intensity of ULK1-757-P normalized for total level of ULK1, PP2A-P normalized for total level of PP2A and p70S6K-P normalized for total level of p70S6K. For each of the experiments, three independent measurements were carried out. Error bars represent standard deviation asterisks indicate statistically significant difference from the control: ns—nonsignificant; *—*p* < 0.05; **—*p* < 0.01. The computational simulation is determined upon (**C**) rapamycin treatment combined with addition of okadaic acid (PP2AT = 0.1, mTORT = 0.1 or (**D**) PP2A silencing combined with rapamycin treatment (PP2AT = 0.01, mTOR = 0.1). The relative activity of mTORC1, PP2A, ULK1 is shown.

**Figure 7 biomolecules-12-01587-f007:**
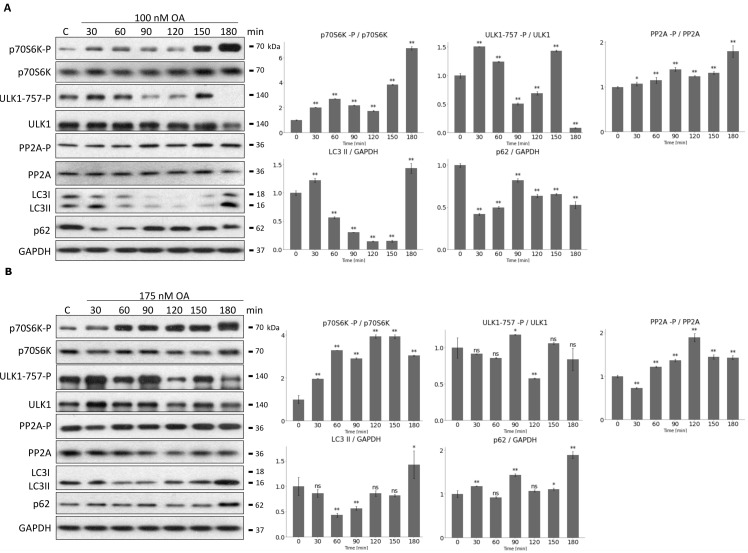
Prolonged OA treatment results in oscillation of PP2A-mTORC1-ULK1 controlled autophagy. HEK293T cells were denoted in time after (**A**) 100 nM or (**B**) 175 nM OA treatment. (**A**,**B**, **panel left**) The markers of ULK1 (ULK1-757-P), PP2A (PP2A-P), and mTORC1 (p70S6K-P) were followed by immunoblotting. GAPDH was used as loading control. (**A**,**B**, **panel right**) Densitometry data represent the intensity of ULK1-757-P normalized for total level of ULK1, PP2A-P normalized for total level of PP2A and p70S6K-P normalized for total level of p70S6K. For each of the experiments, three independent measurements were carried out. Error bars represent standard deviation asterisks indicate statistically significant difference from the control: ns—nonsignificant; *—*p* < 0.05; **—*p* < 0.01.

## Data Availability

Not applicable.

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
