# Peer review of "Fine-Tuning of mTORC1-ULK1-PP2A Regulatory Triangle Is Crucial for Robust Autophagic Response upon Cellular Stress"

_biomolecules, 2022, doi:10.3390/biom12111587_

Round 1

Reviewer 1 Report

In this work by Hajdú et al focused on how the autophagy process occurs and is regulated through the ULK1/PP2A/mTOR1 axis. In this way, these authors reason that two double-negative (which is also known as a positive circuit) between (mTORC1 –| PP2A –| mTORC1, mTORC1 –| ULK1 –| mTORC1) and positive (ULK1 -> PP2A -> ULK1) feedback loops (also known as a circuit) are all essential for the robust, irreversible decision-making process among the autophagy and non-autophagy states. For that, these authors used integration between theoretical and molecular biological techniques. Such as the mathematical systems biology approach along with molecular biology techniques. In addition,  Hajdú et al investigated the dynamical features of the mTORC1- ULK1-PP2A regulatory triangle in detail considering that the positive circuits are crucial to containing a robust cellular answer upon DNA damage. Thus, these authors documented that activated ULK1 can trigger up-regulation of PP2A when mTORC1 is inactivated. By using the mathematical systems biology approach, they describe the significance of cellular PP2A levels in the DNA damage response. They confirmed through the integration between experimentally and theoretically that down-regulation of PP2A (by the addition of okadaic acid) might induce a periodic repetition of the autophagy induction.

Comments:

·       I found this study interesting and relevant in the field.

·   I also found that this study would provide a new layer into the knowledge of autophagy induction by the ULK1/PP2A/mTORC1 axis.

·       The article is well written.

·       The methodology is fine and no further control is required.

·       I found the conclusion to be in line with the evidence and arguments presented.

The manuscript is interesting, however, it can be improved and strengthened by addressing the following comments –

See Figure 1 and the text between lines 143-156, when you explain figure 1. It’s hard to understand: As the authors mentioned It is well-known that there is a double negative feedback loop between mTORC1 and ULK1 kinases (mTORC1 -| ULK1 -| mTORC1). Namely not only the key integrator of intracellular sensing is able to down-regulate the main subunit of autophagy activator complex, but the inducer of the self-digesting process also has a negative effect on mTORC1 via inhibitory phosphorylation (see regulatory connections ‘a’ and ‘b’ on Figure 1)”.

As we can see the Figure 1 (regulatory interaction “a” and “b”), the interaction between mTOR1 à ULK1 and ULK1 --| mTOR1, is that correct?

As the authors noted in Figure 1, between the "A" and "B" double negative feedback loops, is it a double-negative feedback loop?

I think it will be mTOR1 --| ULK1 and ULK1--| mTOR1, am I wrong?

Now, in Section 3.2 and line number 187, the authors found bistable dynamics driven by the double-negative circuits or feedback loops (also known as a positive circuit or positive feedback loop) and positive circuit. The bistability is a combination of autophagy state and physiological state.

We know that positive circuits are responsible for the multistability, i.e, bistability, tristability…etc,  in the network or a system. So, off course, this bistability is controlled by the positive circuit. But the question is which positive circuit is responsible for this bistability?

What is the meaning of the physiological state? I know that it's a non-autophagy state, so it could be an “apoptotic state”?

Why was only one cell line (HEK293T) used to test the hypothesis?

Reviewer 2 Report

This manuscript describes about the regulation of mTORC1-ULK1-PP2A signaling by biochemical and theoretical approaches.  The ideas and discoveries of this manuscript are interesting, but I have found serious concerns about quantification of immunoblot data. 

The appearance of the band intensity does not correlate with its quantified value displayed in the bar graph.  For example, the band of p70S6K-P at 10 min point is almost invisible compared with the control in Figure2A, but the relative value of the p70S6K-P is about 0.25 to the control.  I suppose that the authors may not subtract background signals from each blot.  The authors should reevaluate all the data related to the immunoblot analysis and make discussion on the basis of the revised data.  Also, the authors should describe about the process of quantification in more detail in Method section.

Round 2

Reviewer 2 Report

The authors only corrected the quantification of blots in Figure 2A that I have given as an example in the last comment.  I mean that the authors should reevaluate all the data related to the immunoblot analysis throughout the manuscript.  I think they does not subtract the background from each blot.

Author Response

Reply is attached.
